# Reproductive events and respective faecal androgen metabolite concentrations in captive male roan antelope (*Hippotragus equinus*)

Vanessa W. Kamgang[1]*, Nigel C. Bennett[1], Daniel W. Hart[2‡], Annemieke C. van der Goot[3‡], Andre Ganswindt[1]

1 Mammal Research Institute, University of Pretoria, Pretoria, Gauteng, South Africa, 2 Department of Zoology and Entomology, University of Pretoria, Pretoria, Gauteng, South Africa, 3 Lapalala Wilderness Nature Reserve, Melkrevier, Limpopo, South Africa

☯ These authors contributed equally to this work.
‡ DWH and ACG also contributed equally to this work.
* vanessa.wandja@gmail.com

**Data Availability Statement:** The data generated and analyzed during this study are available on the repository of the University of Pretoria. The digital

## Abstract

Understanding the reproductive biology of the roan antelope (*Hippotragus equinus*) (É. Geoffroy Saint-Hilaire, 1803) is crucial to optimise breeding success in captive breeding programmes of this threatened species. In this study, the pattern of faecal androgen metabolite (fAM) production related to reproductive events (calving or birthing, mating, gestation, and lactation), sexual behaviours as well as environmental cues were studied in captive adult male roan antelope. Faecal sample collection and behavioural observations were carried out from August 2017 to July 2018 for three reproductive males participating in a conservation breeding programme at the Lapalala Wilderness Nature Reserve in South Africa. As a prerequisite, the enzyme immunoassay used in this study was biologically validated for the species by demonstrating a significant difference between fAM concentrations in non-breeding adults, breeding adults and juvenile males. Results revealed that in adults males, the overall mean fAM levels were 73% higher during the breeding period compared to the non-breeding periods, and 85% higher when exclusively compared to the lactation/gestation periods, but only 5.3% higher when compared to the birthing period. Simultaneously, fAM concentrations were lower during the wet season compared to the dry season, increasing with a reduction in photoperiod. With the exception of courtship, frequencies of sexual behaviours monitored changed in accordance with individual mean fAM concentrations in male roan antelope, the findings suggest that androgen production varies with the occurrence of mating activity and may be influenced by photoperiod but not with rainfall.

## Introduction

Cultivation and the associated increase in anthropogenic activity are a global threat to biodiversity [1, 2]. Worldwide, almost a quarter of all mammal species are threatened with extinction [3, 4]. This trend is also reflected in the current populations of African ungulates, which

object identifier assigned to this data is 10.25403/ UPresearchdata.12527414. The data can be assessed following the DOI link https://doi.org/10. 25403/UPresearchdata.12527414.

**Funding:** This work was supported by a DST-NRF SARChI chair for Mammal Behavioural Ecology and Physiology attributed to NCB (Grant number 64756), and a bursary provided by the DST-NRF SARChI chair for Mammal Behavioural Ecology and Physiology to VWK.

**Competing interests:** The authors have declared that no competing interests exist.

have shown extensive declines over the past few decades and are unfortunately projected to decline even more so, especially those species endemic to more arid and semi-arid regions [5–7]. However, mitigation measures such as habitat restoration, translocation and captive breeding appear to be suitable tools for attempting to stabilise vulnerable populations, especially in combination with respective reintroduction programmes [8–10]. To improve captive breeding success, a complete understanding of the biology of an organism is required, with an in-depth knowledge of the reproductive aspects [11]. Thus, many breeding programmes have contributed to a better understanding of the extrinsic and intrinsic factors and requirements necessary for reproductive success, especially by documenting female reproductive physiology and performance [12, 13]. However, for the context of captive breeding programmes, males play as equally an important role as females [14, 15]. Thus, for species participating in conservation breeding programmes, a better understanding of the male reproductive behaviour and related physiology is crucial to optimise breeding efforts for threatened wildlife species. There is a paucity of data focusing on male reproductive physiology in species of the Antilopinae family. Previous studies on male Peninsular pronghorn (*Antilocapra americana peninsularis*) indicated successful copulation occurs during the breeding season with alterations in androgen concentrations following the oestrogen patterns of females [16]. In captive male blackbuck (*Antelope cervicapra*), the levels of androgens are closely linked to aggressive behaviour and scent marking [17].

Androgens are a group of hormones that play a crucial role in the reproductive physiology of males [18]. Across many taxa, androgens including testosterone as a primary male sex hormone, are known to play an essential role in: The activation of spermatogenesis; the development of the reproductive anatomy and the expression and development of secondary sexual characteristics [19]. In addition to these functions, androgens are known to promote male sexual behaviours [20, 21]. The ability to quantify androgens in an increasing number of wildlife species has led to a better understanding of the reproductive physiology and reproductive patterns in many bird and mammal species over the past decades [22–24].

Although utilising blood for hormone analysis is a widely accepted approach for monitoring male reproductive function, non-invasive methods using alternative matrices such as faeces, have gained popularity over the past 45 years [25, 26]. These alternative matrices provide a more practical approach for assessing testicular activity in free-roaming wildlife species [25, 26]. The quantification of faecal androgen metabolites (fAM) has been proven to be extremely valuable for assessing male reproductive function. Its usefulness has been demonstrated in several studies investigating different aspects of the reproductive biology such as proximate cues to the seasonality of reproduction, tusk maturation and challenges associated with reproduction [27–30]. It has been shown that fAM concentrations in male ungulates are correlated to seasonal reproduction, social status [22, 31, 32] and, aggressive and sexual behaviours [33, 34]. Among the Hippotragines, measuring fAM concentrations has been successfully used to evaluate the level of aggression and consequent control in a herd of bachelor fringe-eared oryx (*Oryx gazella callotis*) [34]. Furthermore, this method has been used successfully to monitor and evaluate the success of the introduction of two captive female sable antelope (*Hippotragus niger*) [35]. However, when applied to a species for the first time, respective assays for faecal hormone metabolite monitoring are to be carefully validated to ensure a suitable quantification of respective androgen metabolites [36, 37].

The roan antelope (*Hippotragus equinus*) is a large horse-like antelope with a greyish brown coat and rufous colouring appearance to the face [38]. This antelope is endemic to Africa and inhabits woodlands, grasslands, and savannahs, with numerous populations scattered throughout the western, central, eastern, and southern regions of the continent [39, 40]. Currently, the roan antelope is classified as an aseasonal breeder, as mating and calving have been reported to

occur throughout the year, in both wet and dry seasons with males being reported to breed throughout the year. A breeding herd usually comprises of one dominant reproductive male, multiple females, juveniles, and calves [38, 40]. Males spend the first 30 to 36 months of their existence in the natal herd but are later evicted and become sexually mature at about four years of age [38, 40, 41]. Free-ranging roan antelope populations have experienced a drastic decline in numbers over the past few decades. It is estimated that there are between 50000 to 60000 free-ranging roan antelopes left on the whole continent [42–44]. In southern Africa, roan antelope population numbers have also reduced drastically and they are locally classified as vulnerable [43]. As a consequence of the reduction in numbers, roan antelopes have been bred in captivity for the last 25 years [43, 45]. So far, captive breeding programmes have contributed to the overall increase in roan antelope numbers, although a deeper insight into the reproductive physiology of the species may be helpful to improve the breeding efforts. However, to date, there is a dearth in knowledge on the reproductive physiology, such as the endocrine correlates to reproductive cueing and the timing of breeding for the roan antelope.

As a consequence of the limited knowledge on male reproductive biology in the roan antelope, this study sets out to establish an enzyme immunoassay (EIA) for the monitoring of faecal androgen metabolite (fAM) levels in male roan antelope and subsequently to characterise the pattern of fAM concentrations in males participating in a roan antelope conservation breeding programme. More specifically, the study aims to a) examine the suitability of an enzyme immunoassay to measure fAM concentrations in the male roan antelope, b) relate the profiles of fAM concentrations with monitored reproductive events (calving, mating, lactation and gestation) and the potential roles environmental cues (photoperiod, rainfall and temperature), and c) investigate the relationship between fAM concentrations and the frequency of certain male sexual behaviours displayed during the mating period.

## Methods

### Study area and animals

This study was conducted between August 2017 and July 2018 at the Lapalala Wilderness Nature Reserve (23˚53'3.59"S; 28˚18'0.61"E), situated within the Waterberg Biosphere of the Limpopo province of South Africa. The reserve is a conservation area where roan antelope were reintroduced in 2010 for a continuing captive breeding programme with the aim to successfully reintroduce herds back into the wild. Three captive adult males (7, 8.5 and 10 years; ~ 400–450 kg) were monitored in this study. Each reproductive male was the only adult reproductive male of their respective herd, which otherwise were comprised of adult females, as well as juveniles and calves of both sexes. Each male was either housed in an approximatively 16 -ha camp with its herd mates allowed in "physical contact" (PC) (Fig 1), or was kept alone, in an adjacent camp (8–16 -ha) with "no physical contact" (NPC) with herd mates, but allowing olfactory and visual contact. The camps of the three breeding herds were non-adjacent to each other. All animals grazed on natural vegetation, but were also fed daily with A-grade lucerne (*Medicago sativa)* and protein feed (160 g/kg; 0.9 kg per animal per day) to complement their diet. Water was available *ad libitum*.

This study was carried out without interfering with the actions of the reserve management. Hence, each male was temporarily removed from its respective herd once all the females were assumed to have been impregnated and only returned to the camp when all the adult females had given birth. During this study, the three focal males were reunited twice and removed once from their respective herds and housed in adjacent camps to their respective herd. The entire study was performed with the approval of the University of Pretoria Animal Use and Care Committee (Reference V072-17).

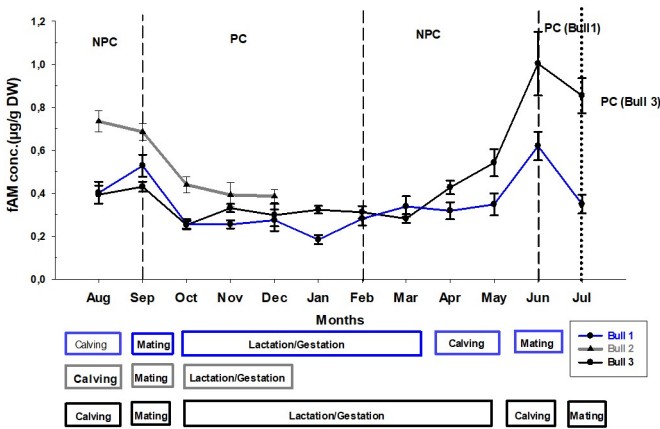

**Fig 1. Monthly variation in fAM concentrations (mean ± SD) in the three male roan antelopes.** PC: physical contact with females, NPC: No physical contact with females. Individual reproductive periods (Mating, calving, and Lactation/Gestation) are indicated (Blue: male 1; Grey: Male 2; Black: Male 3).

## Environmental data

Average monthly temperature (˚C) and cumulative rainfall (mm) at the study site were obtained from the South African weather services (Mokopane station; https://www.weathersa.co.za) for 12 months (August 2017 to July 2018). Additional photoperiod data were obtained from the weather service ClimaTemps (https://www.Climatemps.com), with minutes of daylight per day recorded for the area of Limpopo, South Africa. Seasons were defined as 'wet season,' occurring during the southern hemisphere spring (September to November) and summer (December to February) and as 'dry season', occurring during autumn (March to May) and winter (June to August) [46].

## Behavioural observations

To determine semi-quantitative behavioural data, Male 1 and 2 were observed daily during one of the periods when the males were in physical contact (PC) with their herds (~ 2 months; Male 1: beginning of June until the end of July 2018; Male 2: beginning September till end of October 2017). Daily observations lasted between 2–4 h in total for each male and took place up to three times per day (between 6:30–8:30; 10:30–11:30; and from 14:00–17:00). However, observations were continued whenever behaviours indicated male reproductive activity and stopped latest at sunset. During these observations, faecal samples for hormone analysis were also collected from these two males. All behavioural observations were carried out either on foot or using a vehicle to which animals had become habituated. A minimum distance of 10–30 m was kept from the animals by the observer to avoid disturbance during observations. The occurrence of behaviours related to the male mating activity such as, laufschlag (foreleg kick) anogenital smelling, courtship, flehmen, mounting and courtship circling (mating whirl-around) [41, 47] were recorded using *ad libitum* and continuous recording techniques [47]. Due to logistical restriction, for the rest of the study period, Males 1, and 2 were less intensively observed (Male 1: September 2017 –February 2018; Male 2: November 2017 –December 2017) and observations only took place twice per week during faecal sample collection, lasted 2–3 h in total and focussed on courtship and mounting behaviour. This protocol was also applied for Male 3 which was observed for approximately five months (September 2017 –February 2018 and throughout July 2018) when in PC with the females of its herd. During these observations,

the periods of occurrence of courtship and/or copulatory events were noted to determine the breeding and non-breeding periods. The adult females of the respective herds of each male were also observed to assess their reproductive status (receptive, pregnant, calving or lactating) by monitoring oestrus-related behaviours and behaviours indicating non-receptivity [41].

### Categorisation of reproductive periods

A mating period (breeding period) was defined as the time during which a male was introduced to its respective herd (PC) and ≥70% of all copulations were observed. The successive gestation period was defined as the time during which ≤ 30% of all copulations were observed and females were seen calving after eight months. A calving period was defined as the time between the first and the last calving recorded for a respective herd. A lactation period was defined as the time between the calving and weaning of calves of each herd. Due to the commonly occurring overlap of gestation, calving, and lactation period, these periods were summarised as the non-breeding period (NBP).

### Faecal sample collection

Faecal samples were collected twice weekly; from August 2017—July 2018 for male 1 and 3 and from August 2017—December 2017 for male 2, resulting in a total of 232 samples collected. Once individual defecation was observed, 15–20 g of faeces were collected between 5 to 45 min after the animal had moved away. Samples were collected using gloves while avoiding cross-contamination with urine and other faeces in the area. The collected material was placed in a 30 ml plastic container, then stored in a cooling box containing ice packs; and stored at -20˚C within 4 hours after collection. Samples were kept frozen until further processing.

### Steroid extraction and analysis

Frozen faeces were lyophilised, pulverised and sieved using a thin mesh to remove fibrous material [48]. Following this, 0.100–0.110 g of faecal powder was extracted by adding 3 ml of 80% ethanol. The suspension was subsequently mixed and vortexed (for 15 min) and centrifuged (for 10 min at 1500 x g.). The supernatant was aliquoted in 1.5 ml microcentrifuge tubes and stored at -20˚C until analyses. Faecal extracts were measured for immunoreactive androgen metabolite (fAM) concentrations using an EIA with antibodies against Testosterone-3-CMO:BSA. Detailed EIA characteristics including full descriptions of EIA components and antibody cross-reactivities are described by Palme and Möstl. [49]. Serial dilutions of faecal extracts gave displacement curves that were parallel to the respective standard curve (relative variation (%) of the slope of respective trendlines < 5%). The sensitivity of the assay was 2.4 ng/g dry weight (DW). The intra-assay coefficient of variance (CV) of high- and low-concentration controls were 5.75% and 7.53% respectively. The inter-assay CV of high- and low-concentration controls were 10.65 and 13.77% respectively. All analyses were conducted at the Endocrine Research Laboratory, University of Pretoria, South Africa following Ganswindt et al. [50].

### Biological validation

To determine EIA suitability, the assay was biologically validated by demonstrating its ability to distinguish between male maturation and reproductive stages in terms of immunoreactive fAM concentrations in breeding and non-breeding adults as well as juveniles (<2 years) males. Therefore, additional samples were collected from; five juveniles (one sample per animal) and six non reproductively active males (1–2 samples per animal) at Lapalala Wilderness

Natural reserve. Overall individual median fAM concentrations of reproductively active males (0.60 μg/g DW) were 1.3-fold- and 1.7-fold higher than fAM concentrations of non-reproductively active males (0.27 μg/g DW) and juveniles (0.22 μg/g DW), respectively. Those differences were significant for fAM concentrations between reproductively active males and non-reproductively active males ($t_4$ = 5.42; p<0.001) as well as between reproductively active males and juveniles ($t_4$ = 5.84; p<0.001).

## Data analysis

Faecal androgens metabolite concentrations were assigned according to ecological season (Wet and Dry) and reproductive periods (breeding or non-breeding) and were log-transformed to normalise the data. Subsequently, a linear mixed-effects model approach was used to investigate if the ecological season and/or reproductive periods have an effect on fAM concentrations. In this model, the ID of the animals, the time of the collection (week) were considered as random effects, whereas the ecological seasons and the reproductive events were fixed effects. This model was conducted using lme of the package lme4 package in RStudio (version 3.6.1, R core team 2013).

To examine how fAM concentration influences the display of sexual behaviour in males, the weekly average number of occurrences of reproductive behaviours (mounts, courtships, flehmen, laufshlag, courtship circling between months) for the months of June and July for Male 1 and September and October for Male 2 were determined by dividing the respective total number of recordings by the number of days of observations per week for each month. In addition, weekly individually fAM concentrations for the same months were also determined.

Subsequently, we performed two-tailed Spearman's rank correlation test to examine the relationship between weekly fAM concentrations and weekly number of occurrences of sexual behaviour for male 1 and male 2. Two-tailed Spearman's rank correlation tests were also used to examine how cumulative rainfall, ambient temperatures and photoperiod relate to alterations in fAM concentration.

Apart from the linear mixed-effects models, statistical analyses were carried out using the IBM package Statistical package for social sciences (SPSS) version 25.0 (2019). The results are presented as mean ± standard error (SE) and were found to be significant at p < 0.05.

## Results

### fAM concentrations in relation to ecological season and reproductive period

Overall, a linear mixed-effects model with season (wet vs dry season), reproductive period (breeding vs non-breeding), animal ID, and time of collection (week) as predictors and fAM concentrations as dependent variable showed a significant effect of season (estimate = 0.44, P<0.001) and reproductive period (estimate = 0.54, P<0.001) on fAM concentration.

### Reproductive events and its fAM correlates

At the beginning of the study, the three males were housed individually with no physical contact (NPC) to their females. Once in physical contact (PC), the observed males engaged in courtship activities with females resulting in frequent mounting initially. Most of the breeding activities occurred in September 2017 for all males and again in June 2018 for male 1 and July 2018 for male 3. Following gestation (from September/October 2017, coinciding with lactation), parturition took place in August 2017 (while males were still in NPC) as well during April-June 2018 (Fig 1).

Overall, the analysis of deviance table showed that there is a significant difference in fAM concentration between the reproductive events ($X^2$ = 39.33; df = 2; p<0.001). Individual fAM concentrations (S1 Dataset) increased by 73% during the breeding period (0.59 ± 0.04 μg/g DW; mean ± SD) compared to the entire non-breeding period and by 85% when compared to only the lactation/gestation period only (0.31 ± 0.04 μg/g DW) but increased by 5.28% when compared exclusively to the birth period (0.54 ± 0.11 μg/g DW). Consequently, fAM concentrations during the birth period were 71% higher compared to the lactation/gestation period.

## fAM concentrations in relation to season, temperature, rainfall, and photoperiod

In the dry season (0.54 ± 0.04 μg/g DW), overall individual fAM concentrations increased by 57% compared to the wet season (0.34 ± 0.06 μg/g DW). Results of the analysis of deviance table also showed that there was a significant difference between fAM concentration between the dry and the wet season ($X^2$ = 34.29; df = 1; p<0.001). Furthermore, weekly overall individual fAM concentrations (S2 Dataset) were negatively correlated to temperature (df = 11, r = -0.799, P = 0.002), rainfall (df = 11, r = -0.753, P = 0.01) and photoperiod (df = 11; r = -0.811; P = 0.001) (Fig 2).

## Relationship between fAM concentrations and male reproductive activities

Weekly frequencies (S3 Dataset) of courtship circling (P = 0.05; r = -0.46), anogenital smelling (P = 0.05; r = -0.45) and laufschlag (P = 0.02; r = -0.52) were weakly, but significantly negatively correlated to fAM concentration for both males. A weak but not significant correlation was found between fAM concentration and the frequency of flehmen (P = 0.18; r = -0.32), and the number of mounts (P = 0.4; r = 0.2) and courtships (P = 0.56; r = 0.14) were not correlated. However, profiles of daily fAM concentration and mount frequency followed a similar pattern overall in male 2 which was not evident in male 1 (Fig 3). There was an overall decreasing trend in monthly individual fAM concentrations as well as in the number of recorded male reproductive behaviours (flehmen, courtship circling and mounting) from June to July for male 1 from September to October for male 2, with the exception of the courtship frequency, which increased in July (male 1) and October (male 2) (Table 1).

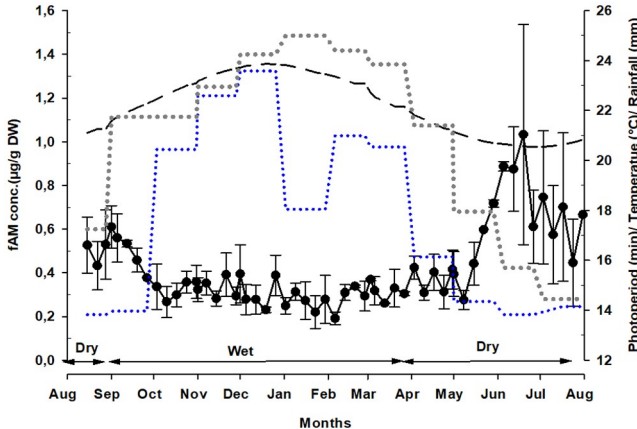

**Fig 2. Weekly overall individual fAM concentrations (μg/g DW) in relation to season, rainfall, temperature and photoperiod across the study period.** The dots and the whiskers represent the mean (± SEM). The dotted line in blue represents the weekly rainfall (mm); the dashed line in black represents the photoperiod (min), the dotted lines in dark grey represent the average weekly ambient temperature. The ecological season is indicated by two-sided arrow lines.

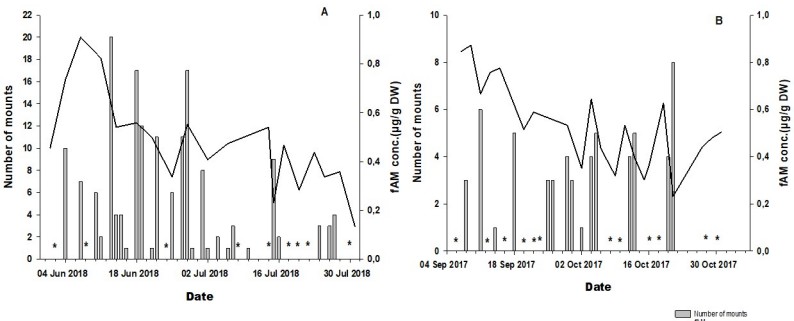

**Fig 3. Daily fAM concentrations (line) and the number of mounts (bars) for male 1 (A) in June and July and for male 2 (B) in September and October.** The stars represent the days of the observation period during which no mounts were recorded.

## Discussion

This is the first study assessing the alterations in faecal androgen metabolite concentrations of captive male roan antelope (*Hippotragus equinus*) in relation to reproductive events, sexual behaviours and environmental cues (photoperiod, temperature, rainfall). Previous studies have shown that male roan antelope adopt a harem breeding strategy where a single male fertilises multiple females of a herd and in this respect, they are similar to other antelopes such as sable (*Hippotragus niger*) and impala (*Aepyceros melampus*) [38, 51].

Male roan antelope fAM levels were elevated during the breeding period and most distinctively during the transition stage of the successive periods of calving and mating. This pattern is in line with findings from other studies on ungulates where male reproductive activity during the rut correlates with an increase in androgen levels [27, 52–55]. For example, androgens levels in captive male bison (*Bison bison*) peak during the rut with concentrations doubling from the pre-rut to the rut period [53]. Similarly, Brown et al. [52] showed that androgen levels increased in free-ranging male impala during the rut. They also showed that androgens levels increase during the breeding period in free-ranging adult male African buffalo (*Syncerus caffer*) [53]. In African elephants (*Loxodonta africana*), faecal androgen metabolite concentrations are distinctively elevated during the occurrence of musth, in mature males [31, 56].

A rise in androgen concentrations is also marked by increased testicular volume and enlargement of the accessory glands [19, 54, 57]; suggesting that androgens play an important role in triggering the physiological processes crucial for the reproductive success of males. Thus the increase of faecal androgen metabolite concentrations prior to actual male mating activity in this study may be related to enhanced spermatogenesis or challenges associated

**Table 1. Monthly fAM concentrations and number of recordings per sexual behaviours recorded (mean ± SEM) across the observational periods for male 1 and 2.**

| Parameters\Animal | Male 1 | | Male 2 | |
|---|---|---|---|---|
| Observation period | June | July | September | October |
| fAM concentration (μg/g DW) | 0.62 ± 0.07 | 0.35 ± 0.04 | 0.69 ± 0.04 | 0.44 ± 0.04 |
| Anogenital smelling | 4.45 ± 0.47 | 3.35 ± 0.63 | 1.86 ± 0.41 | 0.94 ± 0.32 |
| Flehmen | 4.29 ± 0.45 | 3.52 ± 0.61 | 1.14 ± 0.37 | 1.03 ± 0.21 |
| Laufschlag | 7.81 ± 1.38 | 7.81 ± 1.24 | 4.75 ± 0.57 | 3.32 ± 0.70 |
| Courtship circling | 5.50 ± 1.23 | 4.81 ± 0.67 | 4.75 ± 0.57 | 3.32 ± 0.70 |
| Courtship | 1.6 ± 0.19 | 1.83 ± 0.15 | 0.9 ± 0.2 | 1.96 ± 0.28 |
| Mount | 3.81± 0.18 | 1.76 ± 0.08 | 1.39 ± 0.43 | 1.21 ± 0.37 |

with breeding opportunities, such as establishing hierarchies within a herd for species with male-dominance polygyny, such as bison, impala and long-tailed macaque (*Macaca fascularis*) [52, 53, 58].

A pronounced increase in fAM concentrations during the calving period has also been reported in other ungulates and mammals such as non-human primates [24, 28, 58, 59]. In the Peninsular pronghorn, Kersey et al. [16] reported that the rise in androgen concentrations, arising during the prepartum and postnatal periods in males might be a consequence of increased oestrogen levels observed in females, indicating their receptivity to males. This situation may parallel the pattern observed in our study, as an increase in oestrogen levels prior to calving and during the postpartum period, known as foal heat has been shown to occur in other ungulate species [60]. In non-human primate species such as the male verreaux sifika (*Propithecus verreauxi*), similar results have been found; but, Brockman et al. [61] suggest that social disturbance triggered by male movement within groups may be at the origin of the increase in androgens concentrations during the birthing season.

The elevation in fAM concentrations found in the males of our study coincidentally occurs during the dry season, when rainfall, temperatures, and the number hours of light is lower. This is in line with previous findings for captive male goral (*Naemorhedus griseus*), a goat-like antelope, where increasing androgen secretion was also correlated with reduced photoperiod [24]. This pattern of androgen secretion appears to be common during the rut in short-day breeders such as goat and sheep, where androgen concentrations also rise with a regression in day length [62, 63]. Many mammal species are regarded as seasonal breeders [64, 65] and their reproductive activity is usually controlled either extrinsically (environmental variables) or intrinsically (circannual rhythms that are entrained to a particular *zeitgeber* or entrainer) [66, 67]. Day length influences the secretion of melatonin which stimulates the activation of the hypothalomo-pituitary-gonadal axis, thereby affecting the secretion of sex hormones such as androgens [68, 69] in both long and short-day breeders. In goats, for example, the treatment of males with melatonin during a long day photoperiod physiologically stimulated a short day period as it was measured as a long night or short-day and resulted in an increase in the androgen secretion [70].

However, in the tropics, photoperiod is not so important in the regulation of reproductive seasonality. Moreover, it is rainfall as well as subsequent food availability that is important in many African ungulates, such as impala and hartebeest (*Alcelaphus buselaphus*) that exhibit seasonal reproductive patterns that are intimately linked to rainfall [71]. Previous studies on African ungulates have shown that in species such as white rhinoceros (*Ceratotherium simum simum*), the giraffe (*Giraffa camelopardalis*) and the African elephant androgens production may be influenced by various ecological factors [22, 31, 72]. Previous reports on the population ecology of the roan antelope indicate that this antelope is an aseasonal breeder. However, in the Limpopo province and the Waterberg plateau of South Africa where this study was carried out, most of the calving was reported to occur between January and March with a February peak, during the wet season [38, 73, 74]. In roan antelope, faecal androgen metabolite concentrations of the males were, however, lower during the season of rainfall. This pattern goes contrary to studies on other ungulates, which may imply that the captive setting of these antelopes may have had an effect on reproductive seasonality. The animals monitored during this study were moved into breeding camps at a particular time of the year, hence in this study, the occurrence of reproductive activity was influenced by management practices. Furthermore, findings from previous studies reported that captive wild ruminants might not show seasonal reproduction because of the availability of food resources [75], such as in our study with the supplementation of lucerne and protein concentrate. Nevertheless, in the present study, peak fAM concentrations were measured in June/July in two males. Considering that the gestation duration in this species is approximately nine

months, the next calving period is presumed to occur between February and March which is quite similar to the peak calving period in the study area. Hence, there may be periods when breeding is optimal than others, as shown by the increased androgen levels which could lead to increased reproductive success. Moreover, androgens levels increased with reduced day length. Thus, photoperiod could be one of the environmental cues that influencing reproduction in this species as in other mammal species in southern Africa [76, 77].

From a behavioural point of view, our data show that the frequencies of anogenital sniffing, copulation and the flehmen response followed the same pattern as faecal androgen metabolite concentrations, but they were not correlated. These findings are in line with those studies in other Artiodactyla species such as the goral, the Chinese water deer (*Hydropotes inermis*) and pampas deer (*Ozotoceros bezoarticus bezoarticus*) [24, 27, 78]. In male goral, increased androgen levels matched an increase in the frequency of sexual behaviours such as approach, flehmen and mounts [24]. Similarly, for Chinese water deer, high frequencies of smelling and pursuits were recorded concomitantly with heightened androgen levels [78] and in the male pampas deer, androgen concentrations also correlated with reproductive behaviours [27]. However, an unexpected increase in courtship activity was recorded for both males monitored from July and October when comparative faecal androgen metabolite concentrations had decreased. Apart from the low sample size, receptive females might have still been present in the herd during that period and thus may have stimulated courtship behaviour. Moreover, courtship has been observed in both goat and deer that is unrelated to any pattern of androgen secretion [24, 78]. Although there is no direct evidence of the reproductive status of the females in this study, our results may reflect the findings of previous studies on the black rhinoceros (*Diceros bicornis*) where the copulatory activity of the male is strongly influenced by the reproductive status of the female [79]. However, further, investigations should be carried out with a larger sample size to confirm this hypothesis.

## Conclusion

In summary, quantifying androgen metabolites non-invasively has proven to be very useful to monitor male reproductive activity in roan antelope. The results indicate that periods of high fAM concentrations arise during the breeding and birthing periods and can be indicative of the physiological preparation for mating. The observed endocrine pattern also correlates with photoperiod, but seems not to be influenced by rainfall. However further investigations will be necessary as ideally, the study should be repeated over consecutive years to fully interpret the influence of rainfall and photoperiod on the reproductive endocrinology of the male roan antelope. The confirmed relationship of increased fAM levels with heightened reproductive activity may prove helpful to optimise breeding opportunities for this vulnerable antelope that appears to dominate a harem.

## Supporting information

**S1 Dataset. fAM concentrations measured in all males according to weeks, ecological seasons and reproductive events.**
(CSV)

**S2 Dataset. Monthly fAM, photoperiod, temperature and rainfall.**
(CSV)

**S3 Dataset. Average fam, mounts, anogenital smelling, laufschlag, courtship circling per week for male 1 and male 2.**
(CSV)

## Acknowledgments

The authors sincerely thank the Lapalala Wilderness Nature Reserve management, for granting access to their breeding camps, providing data on animals, and providing accommodation to the principal investigator. We also thank staff members of the Lapala Wilderness Nature Reserve for their assistance during the work.

## Author Contributions

**Conceptualization:** Vanessa W. Kamgang, Nigel C. Bennett, Annemieke C. van der Goot, Andre Ganswindt.

**Data curation:** Vanessa W. Kamgang, Daniel W. Hart, Andre Ganswindt.

**Formal analysis:** Vanessa W. Kamgang, Daniel W. Hart, Andre Ganswindt.

**Funding acquisition:** Nigel C. Bennett.

**Investigation:** Vanessa W. Kamgang.

**Methodology:** Vanessa W. Kamgang, Nigel C. Bennett, Andre Ganswindt.

**Supervision:** Nigel C. Bennett, Annemieke C. van der Goot, Andre Ganswindt.

**Validation:** Vanessa W. Kamgang, Nigel C. Bennett, Andre Ganswindt.

**Writing – original draft:** Vanessa W. Kamgang.

**Writing – review & editing:** Vanessa W. Kamgang, Nigel C. Bennett, Daniel W. Hart, Annemieke C. van der Goot, Andre Ganswindt.

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
