## [Decision Letter · Decision Letter 0]

2 Oct 2020

PONE-D-20-19385

Reproductive events and respective faecal androgen metabolite concentrations in captive male roan antelope (Hippotragus equinus)

PLOS ONE

Dear Dr. Wandja Kamgang,

Thank you for submitting your manuscript to PLOS ONE. After careful consideration, we feel that it has merit but does not fully meet PLOS ONE’s publication criteria as it currently stands. Therefore, we invite you to submit a revised version of the manuscript that addresses the points raised during the review process.

I completely agree with the Reviewer's 1 view on seasonality of reproduction can't studied just one year and needs data for two to 3 years. The authors can revise the manuscript as suggested by reviewers and justify critical comments raised by them. Final decision will be taken based on reply  to the comments and revision.

We look forward to receiving your revised manuscript.

Kind regards,

Govindhaswamy Umapathy, PhD

Academic Editor

PLOS ONE

Journal Requirements:

2. Please amend your list of authors on the manuscript to ensure that each author is linked to an affiliation. Authors’ affiliations should reflect the institution where the work was done (if authors moved subsequently, you can also list the new affiliation stating “current affiliation:….” as necessary).

Reviewers' comments:

Reviewer's Responses to Questions

**Comments to the Author**

1. Is the manuscript technically sound, and do the data support the conclusions?

Reviewer #1: No

Reviewer #2: Partly

2. Has the statistical analysis been performed appropriately and rigorously? 

Reviewer #1: Yes

Reviewer #2: Yes

3. Have the authors made all data underlying the findings in their manuscript fully available?

Reviewer #1: Yes

Reviewer #2: Yes

4. Is the manuscript presented in an intelligible fashion and written in standard English?

Reviewer #1: Yes

Reviewer #2: Yes

5. Review Comments to the Author

Reviewer #1: Authors report results about reproductive events and respective fecal androgen metabolite concentrations in captive male roan antelopes. The study was conducted during one year on 3 different males.

The main findings are the validation of the hormonal assay and the increase in androgens during the breeding season.

This is a predictable result.

However, the author claim that androgens levels also were affected by the rainfalls and the photoperiod. The study should be conducted over consecutive years to fully explore those effects.

Behavioral observations also are overinterpreted.

Besides the confirmatory results, it is impossible to draw that many conclusions from this very preliminary set of data.

Reviewer #2: I think that this study has some useful data, but I feel that some clarifications are needed (see below). There are also some errors that require further proofreading.

ABSTRACT

“Results revealed that overall mean fAM levels were 73% higher during the breeding period compared to the non-breeding periods, and 85% higher when exclusively compared to the lactation/gestation periods, but only 5.3% when compared to the birthing period.”

-5.3% higher than during the breeding period? In which group- adult males only?

INTRODUCTION

Lines 49-15: “However, males are equally essential for the context of captive breeding programmes, males play an equally important role as females [14, 15] .”

-I think the wording or grammar here is a bit off.

Line 60: -Though I think it would be obvious to most, perhaps it would be useful to clarify that testosterone is an androgen since the terms are used interchangeably in this paragraph.

Line 71: “faecal andSrogen metabolites”

Line 88: “Currently, the roan antelope is classified as an aseasonal breeder, with a breeding herd comprised of one dominant reproductive male, multiple females, juveniles, and calves”

-Some more information would be useful here, because much of the study is based on understanding this species’ reproduction: Does this mean males breed during both the wet season and dry season? Do they breed repeatedly throughout the year or only once?

When does an adult male qualify as non-breeding? When females are pregnant/calving?

Line 109: “the ptential roles environmental cues”

METHODS

Line 222: “for the months of June and July for Male 1 and September and October for Male 2 were determined by dividing the respective total number of recordings by the number of days of observations per week for each month. In addition, weekly individually fAM concentrations for the same months were also determined.”

-Only 2 males (abstract says 3)? Were fAMs only measured in June-July for male 1 and September-October for male 2? If so, how can seasonal effects and individual effects be separated? When were male 3’s samples collected (if applicable). If I am misunderstanding the study design, please clarify.

RESULTS

Line 246: “…showed a significant effect (Estimate= 0.69, p< 0.001, t=-4.02) of season and reproductive period on fAM concentration.”

-Output for each of the fixed effects should be reported here. Season (estimate and p value) and reproductive period (estimate and p value).

Line 294: “The stars represent the days of the observation during which no mounts was mount were recorded”

-I’m not sure if this is an issue with uploading on the PLOS One website, but figures are extremely pixelated in my PDF and I could not really read them.

DISCUSSION

Line 324: “A pronounced increase in fAM concentrations during the preceding calving period”

-During or preceding?

Line 344: “Day length influences the secretion of melatonin which stimulates the activation of the hypothalomo-pituitary-gonadal axis, thereby affecting the secretion of sex hormones such as

androgens [69, 70] in both long and short-day breeders.”

-If the animals in this study breed year round, this may not be relevant to them

6. PLOS authors have the option to publish the peer review history of their article (what does this mean?). If published, this will include your full peer review and any attached files.

Reviewer #1: No

Reviewer #2: No

---

## [Author Response · Author response to Decision Letter 0]

28 Oct 2020

22 October 2020

We thank both the reviewers for their helpful comments and believe the manuscript has been improved considerably. We have made the necessary changes or responded to each comment where applicable. 

For authors

Ms Vanessa W. Kamgang

General comments

Plos One Reviewer comments

2. Please amend your list of authors on the manuscript to ensure that each author is linked to an affiliation. Authors’ affiliations should reflect the institution where the work was done (if authors moved subsequently, you can also list the new affiliation stating “current affiliation:….” as necessary).

We thank the reviewer for pointing this oversight. Addressed

Comments to the Author

Reviewer #1: Authors report results about reproductive events and respective fecal androgen metabolite concentrations in captive male roan antelopes. The study was conducted during one year on 3 different males.

The main findings are the validation of the hormonal assay and the increase in androgens during the breeding season.

This is a predictable result.

However, the author claim that androgens levels also were affected by the rainfalls and the photoperiod. The study should be conducted over consecutive years to fully explore those effects.

Behavioral observations also are overinterpreted.

Besides the confirmatory results, it is impossible to draw that many conclusions from this very preliminary set of data.

We thank the reviewer for his/her comments. Indeed, we agree with the reviewer, that the study should have been conducted over consecutive years to obtain the complete information on the influence of rainfall and photoperiod on the reproductive endocrinology of the male roan antelope. However, due to logistical and financial constraints, data collection could only be carried out for a year. Even though more investigations may be carried out to demonstrate the effects of environmental season and photoperiod on the secretions of androgens in this species, the results from the current study can be used as a starting point for further research on this topic. We have further discussed the possible influence of photoperiod on the production of androgens in this species (line 505/513) and also emphasized the limitations of this study in the conclusion (line 540/line 543) and recommended that further investigations should be carried out over consecutive years to understand the underlying mechanisms.

Regarding the behavioural results, in this study, we investigated if the variation in androgen metabolite levels affected the display of sexual behaviours in the male roan antelope. As Reviewer 1 mentioned, the findings of this study are preliminary, however, other studies have reported findings similar to those of the present study (line 519/524). Hence, we concluded that the display of behaviours seemed to have been influenced by androgen levels in the male roan antelope. We have now added that further investigations should be carried out with for example / or ideally a larger sample size to support these findings (line 534).

Reviewer #2: I think that this study has some useful data, but I feel that some clarifications are needed (see below). There are also some errors that require further proofreading.

We thank the reviewer for his/her encouraging comments and suggestions to improve the manuscript which contributed to enhancing the quality of the paper.

ABSTRACT

“Results revealed that overall mean fAM levels were 73% higher during the breeding period compared to the non-breeding periods, and 85% higher when exclusively compared to the lactation/gestation periods, but only 5.3% when compared to the birthing period.”

-5.3% higher than during the breeding period? In which group- adult males only?

Yes 5.3% higher, we added “higher” to the statement to improve clarity. In this paragraph we are referring to adult males only and “adult males” has been added to the sentence to improve clarity (line 33).

INTRODUCTION

Lines 49-15: “However, males are equally essential for the context of captive breeding programmes, males play an equally important role as females [14, 15] .”

-I think the wording or grammar here is a bit off.

We thank the reviewer for pointing this out. The sentence was rephrased (line 59).

Line 60: -Though I think it would be obvious to most, perhaps it would be useful to clarify that testosterone is an androgen since the terms are used interchangeably in this paragraph.

We thank the reviewer for pointing this out. The sentence was rephrased (line 70/72).

Line 71: “faecal andSrogen metabolites”

We thank the reviewer for pointing this oversight. Addressed (line 81)

Line 88: “Currently, the roan antelope is classified as an aseasonal breeder, with a breeding herd comprised of one dominant reproductive male, multiple females, juveniles, and calves”

-Some more information would be useful here, because much of the study is based on understanding this species’ reproduction: Does this mean males breed during both the wet season and dry season? Do they breed repeatedly throughout the year or only once?

According to the literature, roan antelope breed throughout the year, with both mating and calving occurring in the wet and dry seasons. These details have been added to the sentence (line 98/100).

When does an adult male qualify as non-breeding? When females are pregnant/calving?

The males were classified as non-breeding during the calving and gestation periods. The respective definitions for the breeding and non-breeding periods are provided in the revised MS at line 185/192. 

Line 109: “the ptential roles environmental cues”

Addressed line 121

METHODS

Line 222: “for the months of June and July for Male 1 and September and October for Male 2 were determined by dividing the respective total number of recordings by the number of days of observations per week for each month. In addition, weekly individually fAM concentrations for the same months were also determined.”

-Only 2 males (abstract says 3)? Were fAMs only measured in June-July for male 1 and September-October for male 2? 

If so, how can seasonal effects and individual effects be separated? When were male 3’s samples collected (if applicable). If I am misunderstanding the study design, please clarify.

We did carry out behavioural observations for the three males. However, in the methodological section, we mentioned in line 173/179 that observations for male 3 observations were less frequent. Due to logistical reasons, we only noted the periods when courtships and copulation occurred to define the breeding and non-breeding periods (line 179/181). 

As mentioned in the methodological section (line 194/195), males 1 and 3 were monitored from August 2017 to July 2018, whereas male 2 was monitored from August 2017 to December 2017. The faecal samples were collected during the study period and the fAM concentrations were measured for each sample collected. This section has been revised for better understanding.

RESULTS

Line 246: “…showed a significant effect (Estimate= 0.69, p< 0.001, t=-4.02) of season and reproductive period on fAM concentration.”

-Output for each of the fixed effects should be reported here. Season (estimate and p value) and reproductive period (estimate and p value).

We thank the reviewer for pointing this out. Respective values of the two estimates have been added accordingly (line 265).

Line 294: “The stars represent the days of the observation during which no mounts was mount were recorded”

We thank the reviewer for pointing this oversight, the term period missing has been added to the sentence (line 316).

-I’m not sure if this is an issue with uploading on the PLOS One website, but figures are extremely pixelated in my PDF and I could not really read them.

We thank the reviewer for pointing this out. The figures have been revised and the high-resolution versions uploaded

DISCUSSION

Line 324: “A pronounced increase in fAM concentrations during the preceding calving period”

-During or preceding?

We thank the reviewer for pointing this oversight, the word preceding has been removed from the sentence (line 349).

Line 344: “Day length influences the secretion of melatonin which stimulates the activation of the hypothalomo-pituitary-gonadal axis, thereby affecting the secretion of sex hormones such as

androgens [69, 70] in both long and short-day breeders.”

-If the animals in this study breed year round, this may not be relevant to them

We agree with the reviewer that if the animals breed throughout the year, one could assume that photoperiod does not influence androgen production. However, we demonstrate peak fAM concentrations for at least two males (1 and 3) in June/July. Therefore, even though these antelopes might breed throughout the year, there may be periods where breeding is preferred, indicated by increased fAM levels, probably facilitating increased reproductive success. Interestingly, previous reports (Joubert 1976, Wilson 1977, Skinner and Chimimba 2005) indicate such periods of high calving rates between January and March in the study area. 

Such adjustments may result in the offspring being born under favourable conditions, and photoperiod could be one of the environmental cues used for identifying the optimal period as for many mammalian species in southern Africa (line 381/402). Hence, we hypothesize that this mechanism could be effective here. However, it could be completely/partially masked by the semi-captive conditions our study males were maintained under.

---

## [Decision Letter · Decision Letter 1]

19 Nov 2020

Reproductive events and respective faecal androgen metabolite concentrations in captive male roan antelope (Hippotragus equinus)

PONE-D-20-19385R1

Dear Dr. Kamgang,

We’re pleased to inform you that your manuscript has been judged scientifically suitable for publication and will be formally accepted for publication once it meets all outstanding technical requirements.

Kind regards,

Govindhaswamy Umapathy, PhD

Academic Editor

PLOS ONE

Additional Editor Comments (optional):

Reviewers' comments:

Reviewer's Responses to Questions

**Comments to the Author**

1. If the authors have adequately addressed your comments raised in a previous round of review and you feel that this manuscript is now acceptable for publication, you may indicate that here to bypass the “Comments to the Author” section, enter your conflict of interest statement in the “Confidential to Editor” section, and submit your "Accept" recommendation.

Reviewer #1: All comments have been addressed

2. Is the manuscript technically sound, and do the data support the conclusions?

Reviewer #1: Yes

3. Has the statistical analysis been performed appropriately and rigorously? 

Reviewer #1: Yes

4. Have the authors made all data underlying the findings in their manuscript fully available?

Reviewer #1: Yes

5. Is the manuscript presented in an intelligible fashion and written in standard English?

Reviewer #1: Yes

6. Review Comments to the Author

Reviewer #1: (No Response)

7. PLOS authors have the option to publish the peer review history of their article (what does this mean?). If published, this will include your full peer review and any attached files.

Reviewer #1: No

---

## [Editor Report · Acceptance letter]

26 Nov 2020

PONE-D-20-19385R1 

Reproductive events and respective faecal androgen metabolite concentrations in captive male roan antelope *(Hippotragus equinus)*  

Dear Dr. Kamgang:

I'm pleased to inform you that your manuscript has been deemed suitable for publication in PLOS ONE. Congratulations! Your manuscript is now with our production department. 

Kind regards, 

on behalf of

Dr. Govindhaswamy Umapathy 

Academic Editor

PLOS ONE